# Recent Advancements in Mosquito-Borne Flavivirus Vaccine Development

**DOI:** 10.3390/v15040813

**Published:** 2023-03-23

**Authors:** Bingan Wu, Zhongtian Qi, Xijing Qian

**Affiliations:** Department of Microbiology, Faculty of Naval Medicine, Naval Medical University, Shanghai 200433, China; wuba@smmu.edu.cn

**Keywords:** flavivirus, mosquito-borne, vaccines

## Abstract

Lately, the global incidence of flavivirus infection has been increasing dramatically and presents formidable challenges for public health systems around the world. Most clinically significant flaviviruses are mosquito-borne, such as the four serotypes of dengue virus, Zika virus, West Nile virus, Japanese encephalitis virus and yellow fever virus. Until now, no effective antiflaviviral drugs are available to fight flaviviral infection; thus, a highly immunogenic vaccine would be the most effective weapon to control the diseases. In recent years, flavivirus vaccine research has made major breakthroughs with several vaccine candidates showing encouraging results in preclinical and clinical trials. This review summarizes the current advancement, safety, efficacy, advantages and disadvantages of vaccines against mosquito-borne flaviviruses posing significant threats to human health.

## 1. Introduction 

Mosquito-borne flaviviruses, members of the genus Flavivirus, are small spherical particles transmitted by various species of mosquitoes [1]. The genome of these viruses is nearly 11 kb in length [2,3,4,5], encoding a single polyprotein precursor which includes three structural proteins, capsid (C), premembrane (prM) and envelope (E), and seven nonstructural (NS) proteins, NS1, NS2A, NS2B, NS3, NS4A, NS4B and NS5) [6,7]. Of these proteins, E protein is found to induce protective immune response, and the significance of antibodies targeting flaviviral E during antiviral protection has been verified in several animal models [8,9,10,11]. Dengue virus (DENV), Zika virus (ZIKV), West Nile virus (WNV), Japanese encephalitis virus (JEV) and yellow fever virus (YFV) are five typical mosquito-borne flaviviruses which could cause severe infectious diseases and, thus, show human medical significance [12]. The wide distribution of accordant mosquito vectors in the world results in the pandemic transmission of these flaviviruses and increases the infection risk of human populations [12,13].

Recently, epidemics of mosquito-borne flavivirus infections pose great concerns [6,14] and are tremendous burdens on global public health systems [15]. DENV affects approximately 130 countries. Nearly 390 million dengue cases are discovered annually [16], of which 96 million (25%) show significant clinical manifestations, with 500,000 cases requiring hospitalization [17,18,19]. DENV causes more human diseases than any other mosquito-borne virus [20], and the number of DENV infected cases has risen over 400% in recent decades [21,22]. ZIKV, first discovered in 1947 in Uganda [23,24,25,26], is a mosquito-transmitted pathogen and is distributed worldwide. It is estimated that about 2.17 billion people are living in regions at risk for ZIKV transmission [27,28,29]. ZIKV transmission can also occur through sexual contact and blood transfusion, or through placental transmission, which causes abnormal gestational development in the fetus [14,27,30,31,32]. Epidemiological investigations have shown that more than 40% of pregnant women infected with ZIKV give birth to fetuses with congenital neurological malformations such as postnatal neurodevelopmental deficits, ocular anomalies, and microcephaly [27,33]. These severe forms of illness place a heavy burden on public health systems [14]. West Nile virus (WNV), a zoonotic pathogen [34], is an important etiological mosquito-borne agent of encephalitis in human and equids [35,36]. Since its first discovery in the last century, WNV has rapidly spread in many countries of the world [37]. In humans, most WNV infected cases are asymptomatic [38,39] or are only present in flu-like symptoms; however, nearly 1% of the infections might develop into more severe neurological diseases, such as encephalitis, meningitis, or flaccid paralysis [38,40,41]. In addition, WNV can be transmitted vertically through the intrauterine route [42], or via breastfeeding milk [43,44] and blood transfusion [45,46,47]. JEV is the most important infectious agent of viral encephalitis in Asia [1,48,49], the endemic regions ranging from the China–Russia border region to northern Australia, and from the Western Pacific islands to the India–Pakistan border [50,51], where an estimated 50,000 clinical cases and 10,000 deaths occur annually [37], with a high incidence of neuropsychiatric sequelae among recovered patients [48,52]. YFV, circulating mainly in Africa and South America [53,54], causes a serious contagious disease, yellow fever (YF). According to the WHO, there are about 200,000 YF cases in clinics annually, with a high fatality rate of about 30% [37].

Although some antivirals, e.g., favipiravir [55], sofosbuvir [56] and TRIM56 [57,58] et al., have been demonstrated to be effective in suppressing some mosquito-borne flavivirus in vitro and in vivo, until now, specific antiviral interventions against these viral infections are not available in clinics [59]. Therefore, effective vaccines represent the optimal strategy to prevent and control these burgeoning diseases. Currently, conventional and experimental mosquito-borne flavivirus vaccines mainly include the following six major categories [60]: (1) live attenuated virus vaccines, (2) inactivated vaccines, (3) nucleic acid vaccines including RNA and DNA [61], (4) virus-like particle (VLP) vaccines, (5) subunit protein vaccines and (6) viral vector vaccines.

This paper will focus on the current status of mosquito-borne flavivirus vaccine development, clarify the composition and efficacy of these vaccines and discuss their advantages and disadvantages.

## 2. Live Attenuated Virus Vaccines

Due to significantly low manufacturing costs [15,62] and comprehensively long-lasting immune response after a single dose [15,59,61,62], live attenuated virus vaccines (LAVs) are the most successful vaccines at preventing infections. An estimated 63% of FDA-approved vaccines are LAVs [59,63]. LAVs against mosquito-transmitted flaviviruses are developing quickly in recent years. YFV-17D is considered to be the safest and most effective LAV against YFV infection [6]. SA14-14-2 is another effective LAV for JEV. One novel LAV technology, ChimeriVax (Chambers et al., St. Louis University Health Sciences Center, USA), has recently manifested considerable prospects in the development of vaccines against mosquito-borne flaviviruses [61]. ChimeriVax is composed of a backbone cDNA clone of a flavivirus, of which the prM-E segment is substituted with the corresponding one from the virus selected for the vaccine. Basically, the backbone comes directly from the virus of LAVs or engineered wild-type (WT) virus attenuated through mutations in vitro [6]. RepliVax (Widman et al., University of Texas Medical Branch, USA) is another emerging technology for the development of flavivirus vaccines based on deletion mutants [64]. These mutant viruses are not capable of assembling and releasing progeny virus particles (single-cycle virus) or replicating their viral genomes [15]. Generally, the flavivirus RepliVax vaccines are constructed by the deletion of C protein from single-cycle viruses, producing subviral particles [15,65] (SVPs, contain E and prM/M protein with antigenically significance, but delete C protein or the viral genome to contribute to the non-infectivity [25,66]). Similarly to ChimeriVax, RepliVax vaccines are relatively safe [15], with the ability to continuously stimulate immune responses [67], and do not require adjuvants, which are used in inactivated virus vaccines (INVs) [64]. However, the reversion of vaccine strains to increased virulence is the main problem related to live virus vaccines. In addition, LAVs are not recommended for the immunocompromised and pregnant women [62].

### 2.1. DENV

A tetravalent recombinant LAV called Dengvaxia^®^ (Sanofi Pasteur, France) [18,68,69] is the only approved DENV vaccine [17,70,71,72,73]. The vaccine is produced by the chimeric technology, in which the prM and E genes of the YFV-17D backbone were substituted by those of DENV 1, 2, 3 and 4 [17,68,69,73,74,75,76]. Studies indicated that Dengvaxia^®^ was safe and induced neutralizing antibodies against DENV of the existing four serotypes [77]. Although Dengvaxia^®^ has been licensed for DENV prevention, global uptake is hampered due to the limitation of only administering it to seropositive persons who are aged above 9 years old, not to mention the sequential 3-dose schedule [21,78]. Takeda Pharmaceutical Company (Tokio, Japan) has developed a chimeric tetravalent LAV candidate (TDV) which contains the attenuated DENV-2-PDK53 strain with the prM and E genes of three other serotypes of DENV [18,71,75,79]. TDV has been confirmed to elicit both long-term humoral [80,81,82,83] and cellular immune response [83,84,85]. A recent phase III clinical trial has shown that TDV was well tolerated [71,86,87,88] and induced 62% seroconversion for four DENV serotypes [18,71]. The primary efficacy data indicated promising results, showing 80.2% overall vaccine efficacy, 95.4% preventive efficacy of severe dengue forms, and 74.9% efficacy in dengue seronegative patients [71]. LATV, another tetravalent LAV for dengue, was developed by the National Institute of Allergy and Infectious Diseases (NIAID) in the US [69,89]. It was composed of four attenuated DENV serotype strains. The phase I–III clinical trials verified that LATV was within an acceptable safety profile [69] and could elicit potent humoral immune responses against all DENV serotypes, with a long-lasting antibody persistence. In addition, the cross-reactive T cell-mediated immune responses are also activated comprehensively by the vaccine [79,88,90,91,92,93]. These data indicate that LATV may serve as an efficient and safe DENV vaccine for application in individuals covering all ages, regardless of DENV infection status before the vaccination [94]. The Walter Reed Army Institute of Research (WRAIR) and GlaxoSmithKline (GSK) have recently codeveloped a tetravalent dengue LAV called TDEN. The TDEN resulted in 100% seroconversion toward all the four DENV serotypes in flavivirus-primed subjects in phase II clinical trials. It has been shown that a two-dose administration regimen was safe and effective in volunteers ranging from 12 months to 50 years of age [69].

### 2.2. ZIKV

ChinZIKV (Beijing Institute of Microbiology and Epidemiology, China) is developed with the JEV SA14-14-2 strain by substituting its prM-E genes with the one from the ZIKV FSS 13025 strain. It has been shown that ChinZIKV protects mice and rhesus macaques against ZIKV challenges, and ZIKV intrauterine transmission was also blocked in pregnant mice [27,30,95,96]. ChimeriVax-Zika is also a chimeric vaccine which contains ZIKV prM-E antigens in the YFV-17D backbone. One dose of the vaccine induced robust ZIKV neutralizing antibodies which protected the mice against ZIKV challenges [27]. Recently, a ZIKV LAV called rZIKV/D4Δ30–713 has entered phase I clinical trial. It is also a chimeric vaccine, which introduces the DENV-4 as the backbone for the delivery of the ZIKV prM-E as the antigens. This chimeric recombinant vaccine was highly attenuated in A129 mice (type I interferon receptor knockout mice) and provided protective immunity against ZIKV challenges [27,96].

### 2.3. WNV

Researchers from the Beijing Institute of Microbiology and Epidemiology have engineered a chimeric JEV/WNV virus (ChinWNV) cDNA which used a subgenomic replicon SA14-14-2 as the genetic backbone, and the prM and E genes of JEV were substituted with the corresponding WNV genes. Studies have shown that one dose of ChinWNV rapidly induced strong humoral immune responses in mice, providing solid protection against the lethal WT-WNV challenge [97]. The US’s NIH has also developed a recombinant WNV LAV (WN/DEN4Δ30), which comprises the prM and E genes of WNV NY99 and the C and NS genes of rDEN4Δ30 [47,98,99]. The results of clinical trials revealed that WN/DEN4Δ30 was safe and immunogenic [98]. Another ChimeriVax WNV vaccine, WNV02 was generated based on the YFV 17D backbone, in which the prM/E genes of the YFV 17D were substituted with those of the WNV NY99 [59,100,101]. Monkeys vaccinated with WNV02 received potent protection against the lethal WT-WNV challenge [15,59] and with high safety [34] and this vaccine has entered phase II clinical trials [102]. A RepliVAX WNV vaccine (RepliVAX WN) containing a large deletion in the C protein gene [66] was demonstrated to provide complete protection against lethal WT-WNV challenges under a single inoculation at the lowest dose in mouse and hamster models [15,66,103].

### 2.4. JEV

SA14-14-2, developed in China [6,13,104], is based on the WT strain SA14, which was attenuated by 114 passages in primary hamster kidney cells [49,105]. Studies indicated that SA14-14-2 was highly immunogenic and a two-dose regimen induced almost 100% seroconversion and protection against the infection [106]. SA14-14-2 is the most widely applied LAV against JEV in the world [49,106] and has been successfully utilized in China with more than 100 million doses administered [105]. However, the theoretical risk for the attenuated SA14-14-2 virus to reverse into a highly virulent strain has, to some extent, limited its extensive application globally to prevent JEV infection [106]. ChimeriVax-JE, a chimeric JEV vaccine, employed YFV 17D as the backbone to express the prM and E of SA14-14-2 as the antigens [6,86,106]. Numerous preclinical studies have suggested that the ChimeriVax-JE vaccine was well tolerated and immunogenic, and that it provides protection against JEV infection in mice and non-human primate (NHP) models. The clinical trials indicated that a single dose of ChimeriVax-JE generated almost the same effect as three doses of JE-VAX (an inactivated JEV vaccine) to produce near-complete seroconversion in the subjects [107,108,109,110]. Ishikawa et al. developed a RepliVAX-JE.2 vaccine based on RepliVAX technology, which expresses the JEV prM and E genes in place of the WNV ones. It was indicated that RepliVAX JE.2 was well tolerated and provided full protection for mice suffered from lethal JEV challenges [65,111,112].

### 2.5. YFV

Yellow fever is the third human infectious disease (after smallpox and rabies) to be controlled by vaccination. A live attenuated YFV (17D), derived from a human isolate (Asibi), has been applied safely and effectively as an LAV for more than 80 years [113], and administered in more than 600 million people around the word [114]. One dose of the vaccine can generate long-lasting specific neutralizing antibodies even for 35 years in some vaccinees, explaining its long-term efficacy [14,114,115]. Another effective YFV LAV (FNV) was produced by attenuating the French viscerotropic strain of YFV in a mouse brain with 128 passages. It was applied in Francophone Africa and showed strong immunogenicity. Unfortunately, FNV was found to increase the neurotropic potential, making it unsafe to utilize in children [14]. FNV was terminated after the last doses were used in 1981 [105,114,116].

## 3. Inactivated Virus Vaccines

Inactivated virus vaccines (INVs) are chemically or physically inactivated whole virions or subunits of viruses [38,117]. The INVs are impossible to revert to a more pathogenic phenotype while containing the whole viral antigens [27], which can induce a balanced antibody response [62,118].The protection of INVs is mainly based on the ability of viral surface proteins, which can induce neutralizing antibodies. The immunogenicity of INVs is greatly enhanced when the antigen is presented in particulate form (virions or SVPs) [119,120]. However, two major disadvantages of INVs include high expense and the need for repeated vaccination [27], both of which make vaccination difficult to achieve in endemic areas [62]. 

### 3.1. DENV

A tetravalent dengue INV was developed by the WRAIR with the GlaxoSmithKline adjuvant systems. This vaccine resulted in robust humoral responses and balanced neutralizing antibody responses for all four DENV serotypes [121].

### 3.2. ZIKV

A purified inactivated ZIKV vaccine, ZPIV, was produced by inactivating the PRVABC59 strain with formalin and by being given an aluminum adjuvant [27,31,96,122]. One dose of ZPIV protected all mice from challenges with the ZIKV-BR strain. Two doses of ZPIV enabled monkeys to remain immune against ZIKV even after a year of vaccination [96,123]. Moreover, the antibodies from immunized monkeys could block ZIKV infection in mice and monkeys in a dose-dependent manner. Currently, ZIPV has entered a phase I clinical trial [27]. The formalin-inactivated MR766 strain was used to develop the ZIKV INV(PIV) [124]. It has been shown that PIV provided full protection against a lethal ZIKV challenge in AG129 mice models after two-dose administration. Importantly, the serum transferred from immunized rabbits also provided protection in mice models [27,96].

### 3.3. WNV

A formalin-inactivated whole-virus veterinary vaccine, WNVAX, (The Research Foundation for Microbial diseases of Osaka University, Japan) derived from the WNV-NY99 strain was licensed in 2003. The efficacy experiments showed WNVAX was a safe and effective vaccine [125,126], and even the 3.2 ng/dose WNVAX could provide 100% protection in the WNV NY99-infected C57BL/6N mice [125]. Another inactivated WNV vaccine, HydroVax-001 WNV, based on the Kunjin strain of WNV (WNV-KV, strain CH16532) [117], was inactivated by 3% H_2_O_2_ [38,47,117]. Vaccination with HydroVax-001 WNV could elicit robust virus-specific neutralizing antibody responses and reduce WNV-associated mortality in BALB/c mice or viremia in rhesus macaques. HydroVax-II, an advanced WNV INV, was inactivated by H_2_O_2_ with Cu^2+^. This INV induced higher (130-fold) WNV-specific neutralizing antibody responses than HydroVax-001 WNV and protected all BALB/c mice from lethal WNV infection [127].

### 3.4. JEV

JE-VAX, a licensed INV for JEV, was produced by inactivating the JEV Nakayama strain with formalin [50,128]. The original virus was replicated in a mouse brain, and the inactive virus was purified by ultracentrifugation [129]. This INV was confirmed to induce robust immune response [130,131,132,133]. However, as an INV, a high cost and the need for repeated administration are still limitations. Moreover, JE-VAX was also associated with some serious allergic and neurologic side effects [134,135,136,137,138]. Production of this vaccine ceased in 2006 due to the above drawbacks, and remaining stocks were depleted in 2011 [139,140]. IC51, an investigational INV against JEV, comprises 6 μg of purified, inactivated SA14-14-2 absorbed to 0.1% aluminum hydroxide [104]. It has been demonstrated that IC51 was considerably immunogenic, highly safe and well tolerated [141]. Two doses of the vaccine induced high protective antibody titers equivalent to three doses of JE-VAX [142]. CVI-JE, a freeze-dried INV, was based on the inactivated Beijing P-3 strain of JEV. The high safety and immunogenicity of CVI-JE has been demonstrated in clinical trials and two doses of the vaccine can induce 100% seroconversion rates [48]. More than seven million doses of CVI-JE have been administered in China since it was registered in 2008 [48,106].

## 4. Nucleic Acid Vaccines

Nucleic acid vaccines generally contain DNA and RNA vaccines [67,75]. DNA vaccines are commonly produced by cloning a promoter and the gene encoding the antigen of the vaccine into a plasmid [27] and they have several advantages over other types of vaccines, including the ability to induce intracellular antigen processing for adaptive immunity [75], the impossibility of the reversion to pathogenic phenotype, stability in extreme temperatures for long periods, and that they are easy to manufacture and low cost [6,62,75,143]. In addition, the DNA vaccines combination is not adversely affected by pre-existing antibodies or the replicative efficiency of each monovalent component [144,145]. However, the possibility of integrating it into the human genome and causing autoimmune diseases makes vaccination with a DNA vaccine a safety risk [6]. Furthermore, the immunogenicity of DNA vaccines is relatively weak in immunized human hosts [143]. Due to the minimal risk of integration into a host genome and a higher safety than DNA vaccines, mRNA technology has been used widely in the development of vaccine for mosquito-borne flavivirus [27,96]. mRNA vaccines utilize the biosynthesis process of host cells to express viral proteins of interest. Moreover, mRNA vaccines are stable in cells due to the untranslated regions at the 5′ and 3′ end, and they can also selectively activate innate immune responses by natural modifications [146,147].

### 4.1. DENV

The U.S. Army Medical Research and Material Command (AMRDCk, WRAIR, NMRC and Vical Inc., Fort Detrick, MD, USA) cloned four plasmids which encode the E and prM of DENV 1, 2, 3 and 4 into the VR1012 plasmid and produced a tetravalent DENV DNA vaccine (TVDV) [148]. The high safety and good tolerability of TVDV has been demonstrated in clinical trials and it induced dose-dependent anti-DENV IFN-γ responses [69]. Importantly, this DNA vaccine can eliminate some viral interference, which has been noted with some DENV LAV_S_ vaccines [75].

### 4.2. ZIKV

The DNA vaccine candidate, termed GLS-5700, was the first Zika vaccine to enter clinical trials [27,31]. The GLS-5700 vaccine was developed by integrating the gene sequences of the prM and E of numerous ZIKV strains [96]. The study suggested that the vaccine could induce high levels of neutralizing antibodies and provide solid protection against ZIKV challenges in mice and NHPs [25]. GLS-5700 was well tolerated in clinical trials and three-dose regimens resulted in high-binding antibody titers to ZIKV-prM/E in all subjects. The antibodies of most volunteers could suppress ZIKV infection in vitro [25]. The other two DNA vaccines, VRC5283 [149] and VRC5288 [150], have been used in NHPs and have shown high ratios of seroconversion. The VRC5283 has been advanced into phase II studies in America [27,31,96].

The ZIKV mRNA vaccines, mRNA-1325 [151] and 1839 [152], were produced by enveloping synthetic RNA molecules with lipid nanoparticles. The experiments have shown that mRNA-1839 provided comparable protection, but higher levels of plasma and memory B cells associated with ZIKV-specific neutralizing antibodies, when compared to DNA vaccine VRC5283. These two vaccines have now entered phase I clinical trials [27]. Self-amplifying mRNA (SAM) is a novel technology for the development of mRNA vaccines [153,154]. The SAM vaccine contains an engineered genome of alphavirus responsible for the replication of vaccines and the genes encoding target antigens [155]. A fever dose of a SAM vaccine can induce stronger immune responses due to the replication characteristic and greater stimulation of the adaptive immune response by the double RNA generated during the replication process [156] than other mRNA vaccines [157].

### 4.3. WNV

A WNV DNA vaccine (pCBWN), containing the genes of viral prM and E, has been shown to result in a strong immune response in horses and full protection in mice when challenged with WNV [158]. Hall et al. developed a plasmid DNA vaccine (pKUN1 plasmid DNA), encoding the full-length RNA of the Kunjin strain. This vaccine was demonstrated to elicit cross-reactive antibodies that could suppress both the New York strain and the Kunjin strain of WNV. The previous results indicated that 0.1–1 μg of pKUN1 plasmid DNA provided solid protection against a lethal challenge with the New York strain of WNV or the Kunjin strain. These data suggest that pKUN1 plasmid DNA may be a promising vaccine candidate to control WNV infection [159].

## 5. Other Types of Vaccines

Other types of mosquito-borne flaviviral vaccines include viral vector vaccines, subunit protein vaccines and virus-like particle (VLP) vaccines. The live viruses, including replication-deficient adenovirus (AV), replication-competent measles virus (MV) and vaccinia virus [27,96], in viral vector vaccines, enable them to infect host cells and result in robust immune responses [160]. Similarly to LAVs, viral vector vaccines cannot be administered to pregnant women or immunocompromised people due to the risk of reversion to increased virulence [27,161]. Subunit protein vaccines are safer to produce and administer than LAVs and INVs since they do not contain any replicative virus [14,37]. Moreover, subunit protein vaccine technology is particularly required for DENV vaccine development as it can activate the immune responses against different serotypes of DENV [37]. Self-assembly properties of viral structural proteins were used to produce VLP vaccines. VLP vaccines contain multiple genes of target viral structure proteins, while lacking the ability of replication [27], which makes them more immunogenic and safe [27,162,163].

### 5.1. DENV

Two complex recombinant AV vector vaccines were produced to express the prM and E proteins of DENV-1, 2 (cAdVaxD12) or DENV-3, 4 (cAdVaxD34). The two vaccines could elicit humoral and cellular immune responses against DENV1-4 from 4 to 10 weeks following primary vaccination [75,143].

For the purpose of avoiding the cold chain and the risk of reversion to pathogenicity, researchers continue to be interested in the study of subunit vaccines [14]. A recombinant protein vaccine for DENV(V180), developed by Hawaii biotech (Merck & Co., Inc., Honolulu, HI, USA), comprised the ectodomains of the E protein of DENV1-4 [18,75]. The genes of viral E protein were amplified by RT-PCR methods and then cloned into pMtt1Xho vector. Studies have indicated that V180 induced neutralizing antibodies to all four DENV and protected rhesus macaques from viremia following the WT-viral challenge [75,164] and clinical trials confirmed that the V180 also induced neutralizing antibodies in volunteers [69,75].

### 5.2. ZIKV

Vesicular stomatitis virus (VSV) is particularly suitable to be developed into a multivalent vaccine due to its compatibility for foreign genes [165,166]. The nature hosts of VSV are livestock; thus, VSV can induce strong systemic immune responses in human populations due to the absence of pre-existing immunity [165,167]. Li et al. developed an attenuated recombinant VSV-based vaccine which generated the prM, E and NS1 of ZIKV. This vaccine candidate induced specific neutralizing antibodies, and T cells mediated immune responses in single-dose immunized mice, which offered full protection against a ZIKV challenge [166]. In addition, AV and MV vector systems are also introduced for ZIKV vaccine development, and are recently entering clinical trials. The AV 26 vector-based vaccine which carries ZIKV M and E (Ad26. ZIKV.001) was demonstrated to possess high immunogenicity. Antibodies from the immunized recipients protected mice against a lethal ZIKV challenge [27,168,169]. A chimpanzee AV system was also applied to generate a ChAdOx1 Zika vaccine, which expressed the prM and E antigen [170]. The efficacy of the vaccine was detected in mice and proved to be 100% protective against ZIKV infection, offering full immunity and reducing viremia and viral dissemination in target organs, such as the brain and ovaries [27]. The AV vectors provide an effective delivery platform due to their minimal immunity in human beings. MV, a live attenuated RNA virus, is among the most secure and efficient human vaccine vectors yet [171]. Two MV vector-based ZIKV vaccine candidates (MV-Zika and MV-ZIKA RSP), which express the prM and E as antigens, are under evaluation in clinical trials. MV-ZIKA RSP was found to be effective in a mouse model to protect the fetus by lowering the viral load during ZIKV infection [172]. A Modified Vaccinia Ankara system (GeoVax) was also utilized to produce a novel vaccine candidate, which delivered ZIKV NS1 as the antigen, perfectly reducing the cross-reacting antibodies and the incidence of antibody-dependent enhancement (ADE) [173]. This vaccine elicited protection against ZIKV challenges in a way that was different from antibody neutralization [27].

To et al. developed a recombinant protein vaccine, which expressed the prM and E of the French Polynesian strain (Accession # KJ776791). The vaccine, in combination with an adjuvant, induced neutralizing antibodies and protected the mice from viremia. [174]. Another recombinant protein vaccine was produced by fusing the E domain III region to the C terminal Fc region of human IgG. Studies have demonstrated that this vaccine could induce strong immune responses and provide solid protection in multiple animal models [175]. More importantly, this vaccine was effective in pregnant CD-1 (ICR) mice models with fetal protection [96,176]. Recently, a ZIKV vaccine expressing the prM, E and NS1 was shown to induce significantly strong immune responses in mice [177].

The first reported vaccine candidate, Zika-VLP, was composed of the constructs that expressed the structural C-prM-E to form the VLP and the nonstructural NS2B-NS3 to catalyze the cleavage of structure proteins, finally producing mature VLP to induce host immunity [178]. The efficacy of the vaccine was verified in mice to induce similar antibody response patterns to the ZIKV INV control. Nevertheless, the amount of specific neutralizing antibodies elicited by the Zika-VLP vaccine was much higher than the ones elicited by INV control. Moreover, no ADE was observed in DENV infection by challenging the Zika-VLP, indicating that dysfunctional responses, such as cross-reacting antibodies induction, were not triggered by the displayed epitopes of Zika-VLP [27]. Immune sera generated from immunized mice were found to protect immunodeficient AG129 mice against ZIKV infection. ZIKVLPs, produced by the University of Wisconsin, was also a vaccine candidate. It could generate potent neutralizing antibodies, reduce viremia in BALB/c mice and elevate the survival rate of AG129 mice after ZIKV challenges [179]. An optimized Zika-VLP vaccine was designated by expressing dimerized E to assemble the VLP of C-prM-E [180]. This optimization helped to develop the envelope dimer epitopes (EDE), which were key to eliciting higher neutralizing antibodies against ZIKV and DENV in mice compared to the VLP with WT sequences [181,182]. The modification was also demonstrated to decrease the ADE risk in in vitro studies [178,183,184]. It was believed that antibodies targeting EDE could prevent viral E from experiencing the conformational change of forming a trimer, thus suppressing viral membrane fusion [185]. VLP-based vaccines are relatively safe due to their inability to replicate in host cells, and will be an ideal option for immunocompromised people or pregnant women [186].

### 5.3. WNV

The recombinant ALVAC^®^-WNV vaccine, based on a modified live recombinant canarypox virus (vCP2017) backbone, expressed the prM/E genes of the WNV NY99 strain, and was formulated in a carbomer adjuvant [59,187]. With the absence of replication in humans and a high attenuation, the vaccine is fairly safe [188]. ALVAC^®^-WNV could elicit WNV-specific neutralizing antibodies in a variety of mammals and provide protection against viremia [189].

MVSchw-sEWNV, a recombinant MV vaccine, expressed the E protein of the WNV IS-98-ST1 strain [171,190]. Vaccination with MVSchw-sEWNV could result in high WNV-specific neutralizing antibody titer and provide protection against a lethal challenge with WNV in CD46-IFNAR mice models. Further experiments indicated that MVSchw-sEWNV was safe and could induce WNV-neutralizing antibodies in squirrel monkeys after a one-dose administration. [190]. Another viral vector vaccine for WNV was developed by cloning the gene sequence of E protein of the WNV LSU-AR01 strain into VSV [191].The vaccine provided 90% protection against the lethal challenge with LSU-AR01 virus in mice models [47].

### 5.4. YFV

With high safety and immunogenicity, YFV-17D is considered to be the most successful mosquito-borne flavivirus vaccine; however, it has been demonstrated that YFV-17D can cause viscerotropic disease and neurotropic disease, which are life-threatening adverse events [192,193,194,195]. Despite the fact that these side effects are rare, there is still a great need to develop safer and more effective vaccines due to the viruses’ high fatality rate [192,196]. The modified vaccinia virus, Ankara (MVA), is one of the most commonly used antigen delivery vectors that is highly attenuated and cannot replicate in humans. Ju lander et al. cloned the genes of YFV PrM and E into an MVA vector and developed YFV vaccine MVA-BN-YF. Studies indicated that MVA-BN-YF induced robust humoral immune response and provided solid protection comparable to that of YFV-17D in YFV-infected hamsters. Importantly, the sera of immunized hamsters can also protect naïve hamsters from lethal infection wth YFV, confirming that high-titer neutralizing antibodies were elicited by MVA-BN-YF. These data suggest MVA-BN YF may represent a safe alternative to YFV-17D [192].

## 6. Discussion

Mosquito-borne flaviviruses, some of the most critical human pathogenic arboviruses worldwide, have seriously affected public health in a number of endemic and/or epidemic regions [197]. These viruses cause a broad spectrum of diseases in humans including fever, encephalitis, meningitis and hemorrhage [198]. In recent years, mosquito-borne flavivirus infections have emerged at an alarming rate worldwide [52]. Effective vaccines are still the best approach to control these diseases due to the absence of effective anti-viral drugs.

During recent decades, breakthroughs have been made in the development of flavivirus vaccines (Table 1). The live attenuated YFV-17D and JEV-SA14-14-2 vaccines are regarded as available, safe and efficient mosquito-borne flavivirus preventatives. YFV-17D, in particular, is considered an excellent model to help develop effective vaccines against flaviviruses [199]. As for the other mosquito-borne flaviviruses, Dengvaxia^®^ has been licensed as a dengue vaccine, and many vaccines (TDV, LATV, TDEV, DPIV and TVDV) for DENV have entered clinical trials (Table 1). There are also vaccines for ZIKV (rZIKV/D4Δ30–713, ZIPV, GLS-5700, MV-Zika, MV-Zika RSP and Ad26.ZIKV.001, VCR5283) and WNV (WN/DEN4Δ30), which have been evaluated in clinical trials (Table 1). Nevertheless, there are no effective vaccines against ZIKV and WNV [37] being used in clinics. YFV and JEV vaccines are already available; however, the ongoing vaccination efforts could hardly prevent infection in risk areas due to a lack of financial resources. 

With the exception of smallpox (for which humans are the only hosts), the eradication of mosquito-borne flaviviruses might never be realized because of the wide distribution of and the difficulty inherent in controlling mosquitos, which means vaccination against mosquito-borne flaviviruses will be a long and continuous process. Moreover, the administration of some effective vaccines has been restricted because of many factors. For example, YFV-17D and SA14-14-2 are not suitable for immunocompromised people and pregnant women, and Dengvaxia^®^ is only recommended to seropositive persons who are above 9 years old [21,78]. These limitations have led to a significant decrease in vaccine coverage. Furthermore, the development of vaccines against mosquito-borne flaviviruses is very difficult on account of some special requirements. For example, one of the most important target groups for the ZIKV vaccine is pregnant women, which creates a unique safety requirement for the vaccine. In addition, the DENV vaccine must focus on tetravalent formulations due to the absence of cross-protection between the four DENV serotypes. Thus, the successful research and development of a mosquito-borne flavivirus vaccine needs to balance various aspects, including the vaccine components, route of vaccination, target population, expense and social financial resources [6]. Some novel technologies, including ChimeriVax, RepliVax, SAM and subunit protein vaccines, have been used as models to construct more effective vaccines and have shown encouraging results in vitro and in vivo, even in clinical trials. These technologies provide promising prospects for the control of mosquito-borne flaviviruses.

## Figures and Tables

**Table 1 viruses-15-00813-t001:** The development and evaluation of mosquito-borne flavivirus vaccines.

Vaccine Types	Virus	Vaccine Candidate	Developer/ Manufacturer	Evaluation	References
LAV	DENV	Dengvaxia^®^	Sanofi Pasteur	Licensed	[17]
		TDV	Takeda Pharmaceutical Company	Phase III (NCT02747927)	[71]
		LATV	NIAID	Phase III (NCT02406729)	[69]
		TDEN	WRAIR and GSK	Phase II (NCT01843621)	[69]
	ZIKV	ChinZIKV	Beijing Institute of Microbiology and Epidemiology	In vivo (animal)	[95]
		ChimeriVax-Zika	Sanofi Pasteur	In vivo (animal)	[27]
		rZIKV/D4Δ30–713	NIAID	Phase I (NCT03611946)	[27]
	WNV	ChinWNV	Beijing Institute of Microbiology and Epidemiology	In vivo (animal)	[97]
		WN/DEN4Δ30	NIH in US	Phase I (NCT00094718 and NCT00537147)	[98]
		ChimeriVax-WN02	Acambis in UK	Phase II (NCT00442169)	[102]
		RepliVAX WN	Department of Microbiology and Immunology in US	In vivo (animal)	[66]
	JEV	SA14-14-2	Chengdu Institute of Biological Products in China	Licensed	[13]
		ChimeriVax-JE	Acambis in UK	Phase III (NCT01092507)	[110]
		RepliVAX-JE.2	Department of Pathology, University of Texas Medical Branch et al.	In vivo (animal)	[112]
	YFV	YFV-17D	Theiler et al.	Licensed	[13]
INV	DENV	DPIV	WRAIR and GSK	Phase I (NCT01666652)	[121]
	ZIKV	ZPIV	NIAD, WRAIR and BIDMC	Phase I (NCT03008122, NCT02963909, NCT02952833 NCT02937233)	[96]
		PIV vaccine	R&D Department Bharat Biotech International Ltd.	In vivo (animal)	[124]
	WNV	WN-VAX	Muraki et al.	In vivo (animal)	[125,126]
		HydroVax-001 WNV	Najít Technologies, Inc, Beaverton, OR in US	In vivo (animal)	[117]
		HydroVax-II	Najít Technologies, Inc, Beaverton, OR in US	In vivo (animal)	[127]
	JEV	JE-VAX	Research Foundation for Microbial Diseases of Osaka University	Licensed	[130]
		IC51	Intercell Biomedical Livingston in UK	Licensed	[106]
		CVI-JE	Liaoning Cheng Da Biotechnology Co., Ltd.	Licensed	[48,106]
Nucleic acid vaccines	DENV	TVDV	AMRDCk, WRAIR, NMRC and Vical Inc. in US	Phase I (NCT01502358)	[148]
	ZIKV	GLS-5700	GeneOne Life Science/Inovio Pharmaceuticals	Phase I (NCT02887482, NCT02809443)	[150]
		VRC5283	NIAID	Phase II (NCT03110770)	[149]
		VRC5288	NIAID	Phase I (NCT02840487)	[149]
		mRNA-1325	Moderna	Phase I (NCT03014089)	[151]
		mRNA-1839	Moderna	Phase I (NCT04064905)	[152]
		SAM	GSK	In vivo (animal)	[153,154]
	WNV	pCBWN	Davis et al.	In vivo (animal)	[158]
		pKUN1 plasmid DNA	Hall et al.	In vivo (animal)	[159]
Viral vector vaccines	DENV	cAdVaxD12/ cAdVaxD34	Raviprakash et al.	In vivo (animal)	[143]
	ZIKV	Ad26.ZIKV.001	Janssen Vaccines	Phase I (NCT03356561)	[168]
		ChAdOx1 Zika	University of Oxford	Phase I (NCT04015648)	[170]
		MV-Zika	Themis Bioscience	Phase I (NCT02996890)	[172]
		MV-Zika RSP	Themis Bioscience	Phase I (NCT04033068)	[27,172]
	WNV	ALVAC^®^-WNV	Minke et al.	In vivo (animal)	[59,188]
	YFV	MVA-BN-YF	Ju lander et al.	In vivo (animal)	[192]
subunit protein vaccines	DENV	V180	Merck & Co., Inc.	Phase I (NCT00936429, NCT01477580)	[69]
	WNV	MVSchwsEWNV	Unite’ des Interactions Mole´culaires Flavivirus-Hoˆtes and Unite´ des Virus Lents in France	In vivo (animal)	[190]
VLP vaccines	ZIKV	ZIKVLPs	University of Wisconsin	In vivo (animal)	[179]
		VLP-cvD	MRC—University of Glasgow Centre for Virus Research	In vivo (animal)	[180]

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
