# Peer review of "Recent Advancements in Mosquito-Borne Flavivirus Vaccine Development"

_viruses, 2023, doi:10.3390/v15040813_

Round 1

Reviewer 1 Report

#minor

1. lines 42-43 "ZIKV transmission can occur sexually, by blood transfusion, or across the placenta to infect the fetus"Introduce in the sentence some word, e.g also, as well, etc  to make clear that these are additional ways of ZIKV transmission beyond the classical mosquito-borne way

2. line 102: use YFV-17D instead of 17D

Author Response

Point 1: "ZIKV transmission can occur sexually, by blood transfusion, or across the placenta to infect the fetus "Introduce in the sentence some word, e.g also, as well, etc. to make clear that these are additional ways of ZIKV transmission beyond the classical mosquito-borne way

Response 1: Thanks for the reviewer’s advice, we have Introduced the word “also” in the sentence (line 42).

Point 2: line 103: use YFV-17D instead of 17D

Response 2: Thank you for your kind suggestions, we have replaced the 17D with YFV-17D (Line 103).

Reviewer 2 Report

This review describes the current state of development of vaccines against dengue, Zika, Japanese encephalitis, West Nile, and yellow fever, all diseases transmitted by mosquitoes and of public health importance.  

STRENGTHS: Comprehensive, informative, updated, and well-written.

WEAKNESSES: Little critic analysis. Add a concluding remarks section with the perspectives of the authors about the field. 

Author Response

Point:This review describes the current state of development of vaccines against dengue, Zika, Japanese encephalitis, West Nile, and yellow fever, all diseases transmitted by mosquitoes and of public health importance.  

STRENGTHS: Comprehensive, informative, updated, and well-written.

WEAKNESSES: Little critic analysis. Add a concluding remarks section with the perspectives of the authors about the field. 

Response:Thanks for the reviewer’s comments and suggestions. We have added our perspectives about the prospects and developments of mosquito-borne flavivirus vaccine in the discussion part (Line436-448 and Line 451-454).
